# Fallacy of attributing the U.S. firearm mortality epidemic to mental health

**Archie Bleyer**[1,2]*, **Stuart E. Siegel**[3], **Jaime Estrada**[4], **Charles R. Thomas, Jr.**[5]

**1** Knight Cancer Institute and Department of Radiation Medicine, Oregon Health & Science University, Portland, Oregon, United States of America, **2** McGovern Medical School, University of Texas, Houston, Texas, United States of America, **3** AYA Cancer Coalition and CureSearch, Los Angeles, California, United States of America, **4** Texas Doctors for Social Responsibility, San Antonio, Texas, United States of America, **5** Radiation Oncology, Geisel School of Medicine @ Dartmouth, Hanover, New Hampshire, United States of America

* ableyer@gmail.com

**Data Availability Statement:** The data and associated analyses are publicly available at https://www.comedsoc.org/2024/02/15/u-s-mental-health-and-firearm-mortality-epidemic/.

## Abstract

### Background

Annual global data on mental disorders prevalence and firearm death rates for 2000–2019, enables the U.S. to be compared with comparable counties for these metrics.

### Methods

The Institute for Health Metrics and Evaluation (IHME) Global Health Burden data were used to compare the prevalence of mental disorders with overall, homicide and suicide firearm death rates including homicides and suicides, in high sociodemographic (SDI) countries.

### Results

Overall and in none of the nine major categories of mental disorders did the U.S. have a statistically-significant higher rate than any of 40 other high SDI countries during 2019, the last year of available data. During the same year, the U.S. had a statistically-significant higher rate of all deaths, homicides, and suicides by firearm (all p<<0.001) than all other 40 high SDI countries. Suicides accounted for most of the firearm death rate differences between the U.S. and other high SDI countries, and yet the prevalence of mental health disorders associated with suicide were not significantly difference between the U.S. and other high SDI countries.

### Conclusion

Mental disorder prevalence in the U.S. is similar in all major categories to its 40 comparable sociodemographic countries, including mental health disorders primarily associated with suicide. It cannot therefore explain the country's strikingly higher firearm death rate, including suicide. Reducing firearm prevalence, which is correlated with the country's firearm death rate, is a logical solution that has been applied by other countries.

**Funding:** The author(s) received no specific funding for this work.

**Competing interests:** The authors have declared that no competing interests exist.

## Introduction

The firearm injury and death rates in the U.S. are at an all-time high and continuing to increase. Its gun lobby blames the crisis on citizens with mental health disorders and not on firearm prevalence. The National Rifle Association, which arguably wields far greater influence over national firearms policy than does public opinion [1], lays the blame for mass shootings on untreated mental illness—rather than unregulated guns—and proposed the creation of a national database of persons with mental illness [2]. Indeed, the NRA's mantra is "The only guy that can stop a bad guy with a gun is a good guy with a gun" [3]. Recent availability of data on firearm homicides, suicides and unintentional deaths and on mental disorders in the world's countries and territories [4] allows testing of the hypothesis that the country's mental health is a primary reason for its firearm mortality crisis.

## Methods

The mean and 95% confidence intervals (C.I.) of mental health disorders prevalence were obtained for the U.S. and other high sociodemographic index (SDI) [5] countries from the Institute of Health Metrics and Evaluation (IHME) Global Health Burden resource [4]. For each of nine mental health disorders provided by IHME, the U.S. was compared with each of 40 other high SDI countries for overall firearm deaths, firearm homicides, and firearm suicides. Annual firearm death rate data in the U.S. was obtained from the Centers for Disease Control and Prevention Web-based Injury Statistics Query, Reporting and WONDER Systems [6,7]. U.S. data were available until 2021 whereas IHME SDI country data were not available after 2019. The Joinpoint Regression Program, version 5.0.1-April 2023 [8] and its constant variance option, was used to identify mortality trends and when they occurred. Joinpoint-derived p-values in the figures are designated in the text as <<0.001 if several logs <0.001.

## Results

### Mental disorder prevalence

In none of the nine major categories of mental disorders provided by IHME did the U.S. have a statistically-significantly higher rate than any of the other high SDI countries during 2019, the last year of available data (Fig 1). In rank order of the 41 high SDI countries, the U.S. was 5th in all mental health disorders and 12th, 2nd, 4th, 32nd, 35th, 1st, 4th, 21st, and 9th in anxiety, depression, attention deficit/hyperactive, bipolar, conduct, schizophrenia, autism, eating, and idiopathic developmental intellectual disability disorders, respectively (Fig 1).

### Firearm death rate

During the last two decades, the U.S. had rate increases since 2006 for firearm suicides and since 2010 for all firearm deaths and firearm homicides (all p<<0.001) (Fig 2). In contrast, the average rate in other high SDI countries decreased during 2000–2019 for all firearm deaths, firearm suicides and firearm homicides (p = 0.003 for all firearm deaths) (Fig 2).

In both the U.S. and other high SDI countries the majority of firearm deaths during 2010–2019 were suicides (59% and 62%, respectively) (Fig 2). In comparison with suicides by specified methods other than firearm, the U.S. had a greater increase in firearm suicides than did other high SDI countries (Fig 3 left panels). For every year during 2000–2019, the percentage of all suicides that occurred by firearm was >50% in the U.S. and <10% on average in other high SDI countries. Each of the other high SDI countries had decreases in the firearm suicide rate, especially Canada, Switzerland, Finland, France, France, Belgium, Slovenia, Norway, Estonia and Latvia and only San Marino and Monaco did not have substantial decreases from

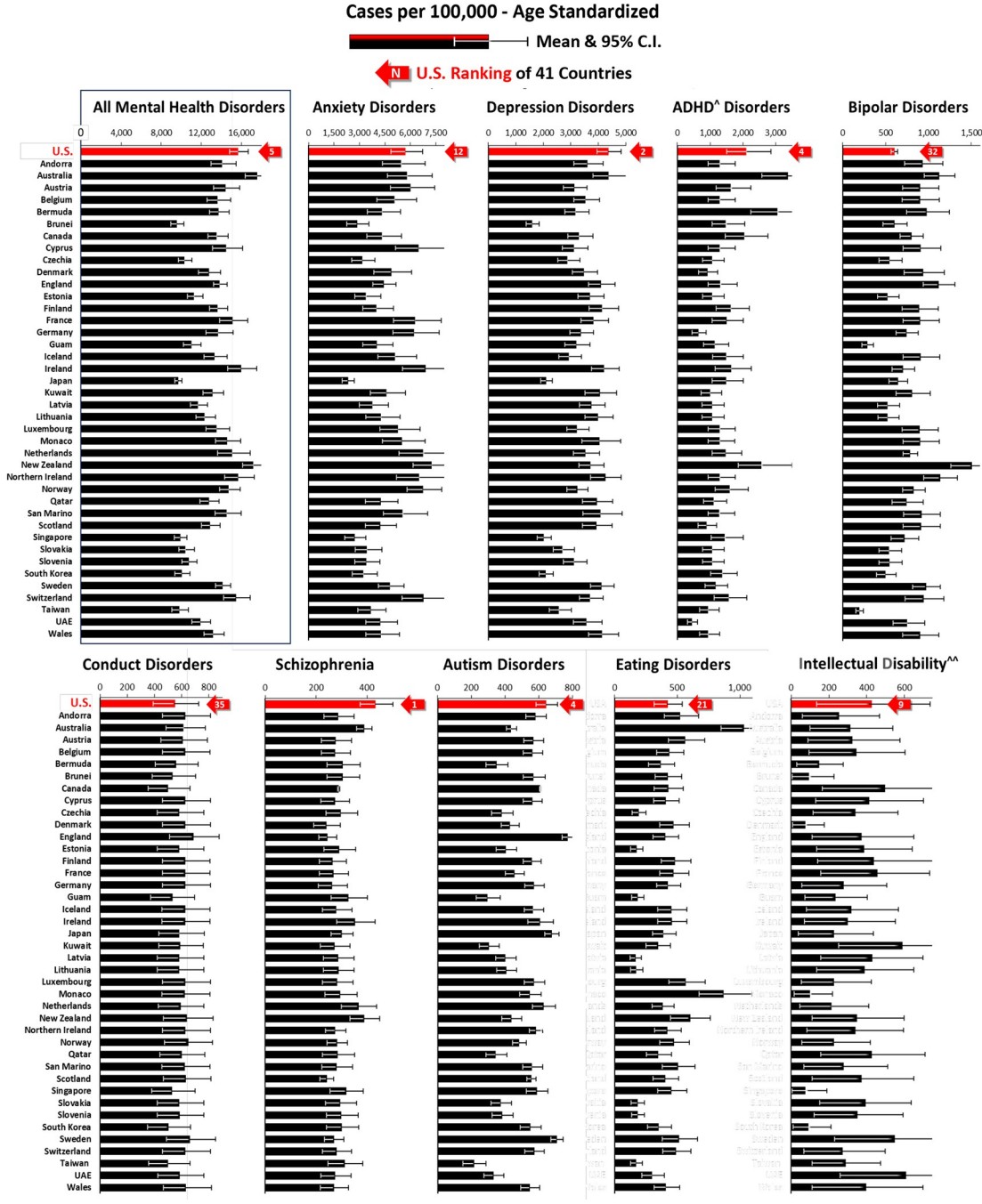

**Fig 1. Mean & 95% C.I. of mental disorders prevalence in 41 high sociodemographic index (SDI) countries, 2019, by disorder type.** Upper panel: Anxiety, Depression, Attention Deficit/Hyperactivity Disorders, and Bipolar Disorders. Lower panel: Conduct Disorders, Schizophrenia, Autism Disorders, and Other Mental Disorders. *SDI- Sociodemographic Index. **ADHD—Attention Deficit/Hyperactivity Disorders. Data Source: IHME [4].

relatively higher rates (Fig 3 upper right panel). For non-firearm suicide, the U.S. had an increasing rate during 2000–2019, whereas the rate significantly decreased overall in high SDI countries and only Guam had an increase among all other high SDI countries (Fig 3 lower left panel). As of 2019, all but 8 of the other 40 high SDI countries had a greater rate than the U.S. (Fig 3 lower right panel)

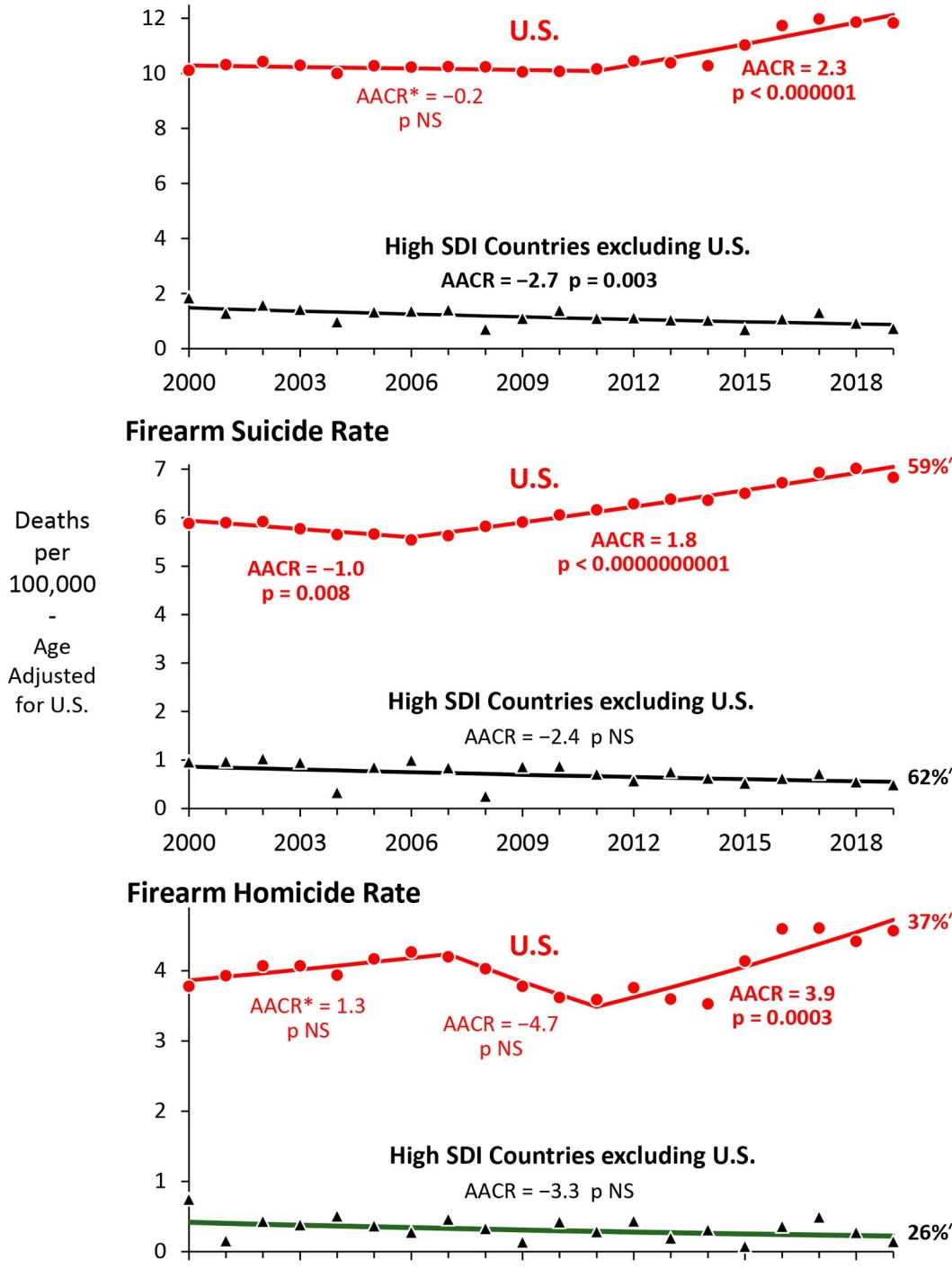

**Fig 2. Joinpoint/AAPC\* analysis of annual overall firearm death, homicide and suicide rates, 2000–2019, U.S and other high SDI countries.** \*AAPC–Average Annual Percent Change. Data Sources: CDC WISQARS [6] for U.S.; IHME [4] for other high SDI countries.

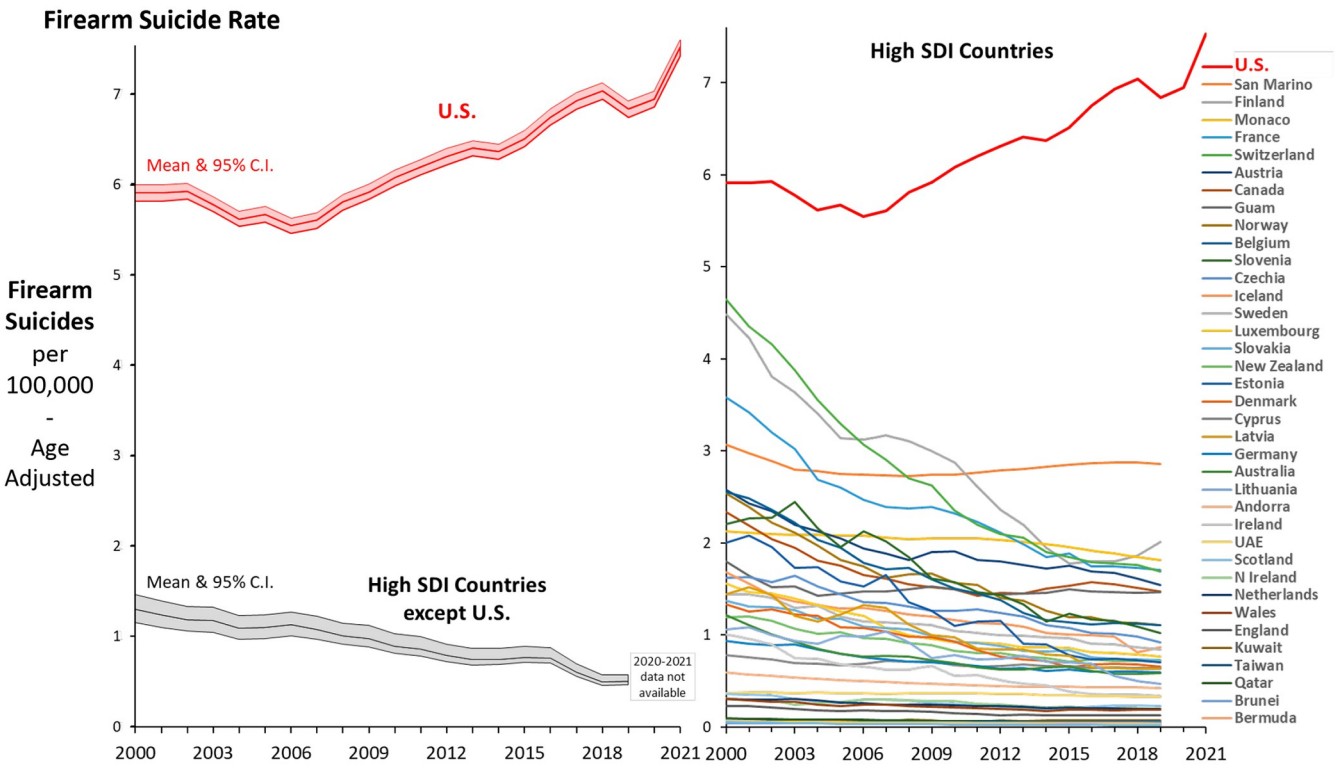

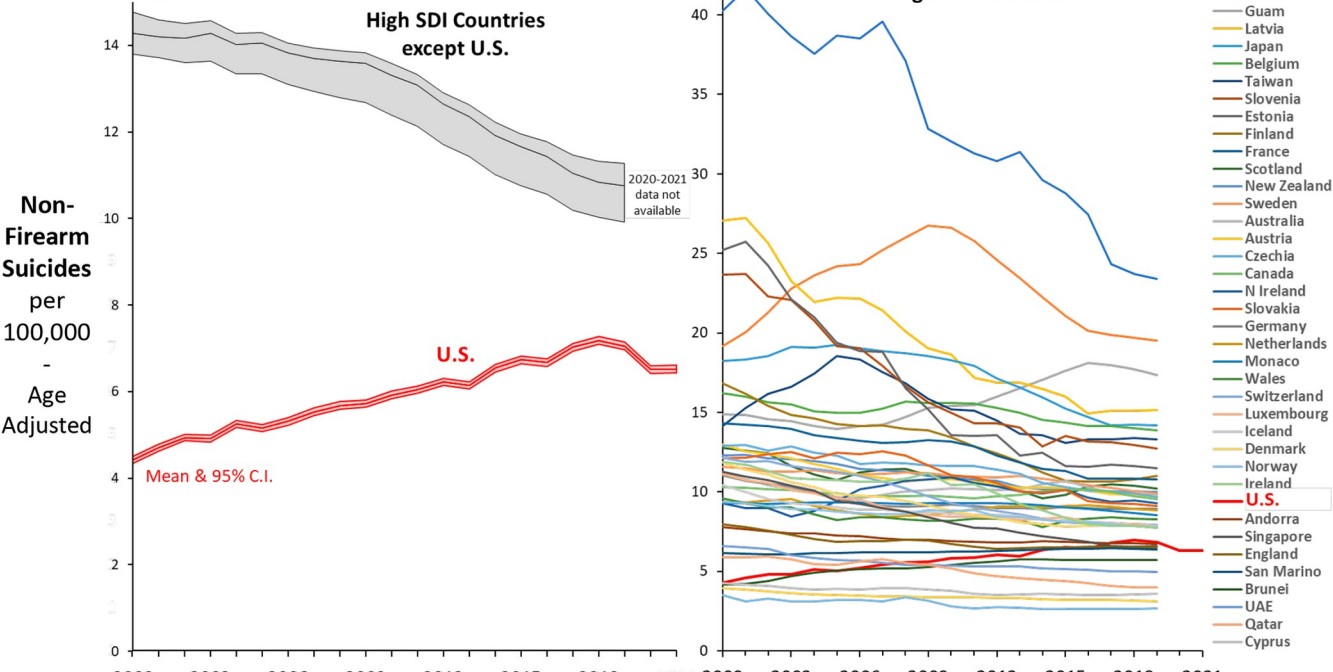

**Fig 3. Annual suicide rate by firearm (upper panel) and by method other than by firearm (lower panel), U.S. 2000–2021 and other evaluable high SDI countries 2000–2019.** Data Sources: CDC WONDER [7] for U.S., IHME [4] for other high SDI countries.

In 2019, the U.S. had a statistically-significant higher rate of overall firearm deaths, firearm suicides, and firearm homicides than each of the other 40 high SDI countries (all 120 p-value differences <0.001) (Fig 4). Compared with the average of all other high SDI countries, the U.S. rate was 10.1 times higher for overall firearm, 7.9 times higher for firearm suicide, and 18.6 times greater for firearm homicides.

During the last decade, the age-adjusted overall firearm death rate increased statistically significantly among the high SDI countries in only the U.S. (AAPC = 2.2, p<0.001) and, at a much lower rate, San Marino (AAPC = 0.5, p<0.001) (S1 Fig). Of the 39 other high SDI countries, 33 had a statistically-significant decrease and 6 had no significant change (S1 Fig).

## Comparisons of firearm death rate and mental health disorder prevalence

Of 150 comparisons of overall, homicide and suicide firearm death trends with mental health disorder trends evaluated for 2000-2019 eras in the U.S., only 10 were either strongly or highly correlated (Pearson correlation coefficients >0.80) (Table 1). Firearm suicide was correlated with conduct disorders for eras during 2000-2019, with autism disorders during 2005-2019, and attention deficit and eating disorders during 2015–2019, and these correlations were reflected in the overall firearm death correlations (Table 1). None of the firearm homicide comparisons were correlated (Table 1).

## Discussion

Our results indicate that the U.S. has a similar prevalence of mental disorders to that in 40 other high SDI countries. In 2019, the U.S. had an overall mental health disorder ranking of 5th and average ranking of 13 among the nine categories of mental disorders among the 40 high SDI countries. Its firearm death rate ranking, however, is by far in 1st place, 10-fold greater in 2019 than the average of all other high SDI countries for all firearm deaths and 19-fold for firearm homicides. Suicides accounted for more of the firearm death rate differences between the U.S. and other high SDI countries than homicides, and yet the prevalence of mental health disorders associated with suicide such as depression, anxiety, schizophrenia, bipolar and conduct disorders, were not significantly difference between the U.S. and other high SDI countries. Our results include lack of correlation of the U.S. firearm mortality trends with its mental disorder prevalence trends, with possible exceptions of conduct, autism, attention deficit and eating disorders correlated only with firearm suicide and limited to certain eras since 2000.

The difference between the U.S. and all other high SDI countries cannot be explained by differences in mental disorder prevalence. Yet, a significant proportion of the U.S. populace attributes its firearm mortality and injury epidemic to inadequate care of its mentally-disabled population. A decade ago, a public opinion poll found that a majority of Americans across the political spectrum favored "increasing government spending to improve mental health screening and treatment as a strategy to prevent gun violence" [9]. As of last year, a national poll found that nearly half of respondents believe mass shootings are more common in the U.S. than in other countries because of mental health issues, and that a majority of both Republicans and Democrats respondents believe better mental health screening and treatment would be one of the most effective ways to prevent mass shootings [10]. Mental health experts and consumer advocates thus face the difficult prospect of debunking the public perception that "the mentally ill are dangerous" [11].

Multiple studies have correlated the U.S. firearm mortality and injury rates with firearm prevalence as estimated by the firearm background check rate [12–16], the proportion of suicides by firearm [13,16–25], surveys [26–29], and the proportion of the world's firearms in

**Fig 4. Firearm death rates, high SDI countries, 2019, by country and type of firearm death.** Data Source: IHME [4].

**Table 1. Pearson correlation coefficients (r) of comparisons of age-adjusted firearm death rates and age-adjusted prevalence of mental disorders, 1990–2019, by type of firearm death, mental disorder, and Era, U.S. Bolded values: r > 0.80.**

| Era: | 1990–2019 | 1995–2019 | 2000–2019 | 2005–2019 | 2010–2019 | 2015–2019 |
|---|---|---|---|---|---|---|
| | | | **All Firearm Death Rate** | | | |
| All Mental Disorders | -0.83 | -0.65 | -0.63 | -0.66 | -0.50 | 0.35 |
| Anxiety Disorders | -0.63 | -0.55 | -0.58 | -0.61 | -0.42 | 0.31 |
| Depressive Disorders | -0.92 | -0.93 | -0.89 | -0.89 | -0.86 | -0.90 |
| ADHD* | -0.82 | -0.41 | 0.50 | 0.49 | 0.69 | **0.95** |
| Bipolar Disorder | 0.64 | 0.14 | -0.67 | -0.75 | -0.96 | -0.93 |
| Conduct Disorder | 0.07 | 0.31 | **0.84** | **0.83** | 0.78 | -0.77 |
| Schizophrenia | -0.61 | -0.66 | -0.92 | -0.94 | -0.95 | -0.97 |
| Autism Disorders** | -0.35 | 0.01 | 0.55 | 0.50 | 0.77 | 0.61 |
| Eating Disorders | -0.49 | 0.04 | -0.78 | -0.77 | -0.67 | **0.94** |
| Intellectual Disability*** | 0.49 | -0.36 | -0.64 | -0.67 | -0.86 | -0.61 |
| | | | **Firearm Homicide Rate** | | | |
| All Mental Disorders | -0.75 | -0.45 | -0.25 | -0.21 | -0.44 | 0.28 |
| Anxiety Disorders | -0.53 | -0.34 | -0.20 | -0.14 | -0.37 | 0.25 |
| Depressive Disorders | -0.88 | -0.77 | -0.55 | -0.55 | -0.80 | -0.73 |
| ADHD* | -0.81 | -0.40 | 0.63 | 0.66 | 0.68 | 0.75 |
| Bipolar Disorder | 0.71 | 0.35 | -0.25 | -0.23 | -0.90 | -0.69 |
| Conduct Disorder | -0.07 | 0.03 | 0.43 | 0.42 | 0.72 | -0.57 |
| Schizophrenia | -0.47 | -0.40 | -0.56 | -0.59 | -0.89 | -0.76 |
| Autism Disorders** | -0.47 | -0.27 | 0.05 | -0.07 | 0.72 | 0.44 |
| Eating Disorders | -0.36 | 0.28 | -0.39 | -0.38 | -0.61 | 0.70 |
| Intellectual Disability*** | 0.59 | -0.09 | -0.19 | -0.13 | -0.80 | -0.45 |
| | | | **Firearm Suicide Rate** | | | |
| All Mental Disorders | -0.89 | -0.79 | -0.78 | -0.84 | -0.57 | 0.33 |
| Anxiety Disorders | -0.77 | -0.73 | -0.74 | -0.83 | -0.49 | 0.29 |
| Depressive Disorders | -0.85 | -0.91 | -0.90 | -0.90 | -0.90 | -0.90 |
| ADHD* | -0.70 | -0.23 | 0.27 | 0.19 | 0.66 | **0.95** |
| Bipolar Disorder | 0.36 | -0.24 | -0.83 | -0.96 | -0.99 | -0.99 |
| Conduct Disorder | 0.40 | 0.64 | **0.93** | **0.92** | **0.84** | -0.80 |
| Schizophrenia | -0.81 | -0.85 | -0.92 | -0.95 | -0.98 | -0.97 |
| Autism Disorders** | -0.01 | 0.43 | 0.79 | **0.83** | 0.77 | 0.64 |
| Eating Disorders | -0.70 | -0.36 | -0.86 | -0.86 | -0.74 | **0.98** |
| Intellectual Disability*** | 0.18 | -0.66 | -0.84 | -0.92 | -0.87 | -0.62 |

\* ADHD -Attention-deficit/hyperactivity disorder.

\** Autism spectrum disorders.

\*** Idiopathic developmental intellectual disability.

civilian hands [30]. As reported recently in the American Association of Medical Colleges News, focusing on mental illness as the cause of firearm violence diverts attention from the larger problem of gun violence in the U.S., and distracts from the real issue when it comes to guns and mental health: suicide [31]. The strikingly higher suicide rate in the U.S. has been attributed primarily to access of suicidal persons to firearms that exists in few in any other countries. Other high SDI countries have decreased firearm prevalence and subsequently reduced their firearm death rate [32], including Australia [33], Canada [34,35], New Zealand [36], Switzerland [27,37], and Israel [37] (S2 Fig). According to a comparison of Canada with

the U.S., 1 in 4 U.S. suicide fatalities could be averted if the U.S. had the same suicide rates as in Canada and its lower firearm ownership rates [38].

Although the primary limitation of this study is its ecologic nature, case control and cohort studies support firearm access *per se* as a causal factor of firearm mortality and not mental health *per se*. In the U.S., higher rates of firearm ownership at the state level have been shown to be strongly associated with higher rates of firearm suicide but not with non-firearm suicide or gun ownership level [39]. The authors concluded that firearm ownership rates independent of underlying rates of suicidal behavior largely determine variations in suicide mortality across the 50 states [39]. In the U.S., a nationally representative study of 10,123 13–18 year-olds estimated that their risk of suicide was increased 3–4 times if they had lived in homes with a firearm compared with if they had not [40]. An analysis of four U.S. studies did not suggest any other confounders that explain the association between firearms and suicide [41].

Other examples have evidenced that suicide rates can be substantially reduced without targeting underlying mental health or suicidality. The success in preventing suicides in Sri Lanka by reducing access to the most highly toxic pesticides is, as the authors conclude, "one of the strongest empirical arguments" and supports 'household firearm ownership as a consistent strong predictor of suicide risk in studies that examined individual-level data" [42].

None of this minimizes the need for the U.S. to provide more support, especially research and including funding, for its overall mental health. As our study shows, the U.S. is also among high SDI countries with the greatest prevalence for depression, schizophrenia, attention deficit/hyperactivity and autism disorders. The trends in the U.S. of firearm death rates do not correlate, however, with either its overall mental disorder prevalence trend or with any of the major mental disorder types except conduct disorder. The relative ranking among the high SDI counties could be due in part to the psychosocial impact of the firearm injury and mortality epidemic in the U.S., rather than to the reverse of mental disorder prevalence causing the country's firearm crisis.

With the onset of the Covid-19 pandemic in 2020, the U.S. has had acceleration of firearm purchases [43] and firearm deaths [44]. As recently reported in PLOS-ONE, pandemic gun buyers surveyed exhibited more mental health characteristic, including suicidality, depression, anxiety, and substance use than non-gun owners and pre-pandemic gun owners [45]. As of 2019 and the prior two decades, however, mental health disorder rates in the U.S. do not explain the country's firearm mortality crisis.

In summary, mental disorder prevalence in the U.S. is similar in all major categories to its 40 comparable sociodemographic countries, including mental health disorders primarily associated with suicide. The country's firearm crisis should not be blamed on lack of mental healthcare. Reducing firearm prevalence, which is correlated with the country's firearm death rate, is a logical solution that has been applied successfully by other countries.

## Supporting information

**S1 Fig.**
(PDF)

**S2 Fig.**
(PDF)

**S1 File.**
(PDF)

**S2 File.**
(PDF)

**S3 File.**
(PDF)

**S4 File.**
(PDF)

**S5 File.**
(PDF)

## Author Contributions

**Conceptualization:** Archie Bleyer, Stuart E. Siegel, Charles R. Thomas, Jr.

**Formal analysis:** Archie Bleyer.

**Investigation:** Archie Bleyer.

**Methodology:** Archie Bleyer, Stuart E. Siegel.

**Resources:** Charles R. Thomas, Jr.

**Writing – original draft:** Archie Bleyer.

**Writing – review & editing:** Stuart E. Siegel, Jaime Estrada, Charles R. Thomas, Jr.

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
