## [Decision Letter · Decision Letter 0]

1 Dec 2023

PONE-D-23-22215Fallacy of Attributing the U.S. Firearm Mortality Epidemic to Mental HealthPLOS ONE

Dear Dr. Bleyer,

Thank you for submitting your manuscript to PLOS ONE. After careful consideration, we feel that it has merit but does not fully meet PLOS ONE’s publication criteria as it currently stands. Therefore, we invite you to submit a revised version of the manuscript that addresses the points raised during the review process.

The authors must focus on integrating more peer-reviewed literature in the introduction and discussion.

We look forward to receiving your revised manuscript.

Kind regards,

Claudio Alberto Dávila-Cervantes, Ph.D.

Academic Editor

PLOS ONE

Journal Requirements:

2. We notice that your supplementary figures are included in the manuscript file. Please remove them and upload them with the file type 'Supporting Information'. Please ensure that each Supporting Information file has a legend listed in the manuscript after the references list.

Reviewers' comments:

Reviewer's Responses to Questions

**Comments to the Author**

1. Is the manuscript technically sound, and do the data support the conclusions?

Reviewer #1: Yes

Reviewer #2: Yes

Reviewer #3: Yes

2. Has the statistical analysis been performed appropriately and rigorously? 

Reviewer #1: Yes

Reviewer #2: Yes

Reviewer #3: Yes

3. Have the authors made all data underlying the findings in their manuscript fully available?

Reviewer #1: Yes

Reviewer #2: Yes

Reviewer #3: Yes

4. Is the manuscript presented in an intelligible fashion and written in standard English?

Reviewer #1: Yes

Reviewer #2: Yes

Reviewer #3: Yes

5. Review Comments to the Author

Reviewer #1: The authors deserve my sincere congratulations. It has been quite some time since I reviewed such a good paper.

The introduction is straightforward , the hypothesis is highly relevant, the data source is reliable, the statistical analysis is simple but efficient, and the discussion is clear and concise.

Reviewer #2: The manuscript is sound and addresses an important topic. Statistical analysis is also valid. However, it lacks a literature review on the topic. Authors should also rephrase some sentences to incorporate passive voice only.

Reviewer #3: The manuscript under review, “Fallacy of Attributing the U.S. Firearm Mortality Epidemic to Mental Health” is a cogent empirical essay that underscores an observation in need of reinforcement. Namely, that mental disorder prevalence in the U.S. is similar in all major categories to its 40 comparable sociodemographic countries and cannot explain the U.S. firearm crisis. None of this minimizes the need for the U.S. to support overall mental health, as the authors take care to note.

The paper could be strengthened substantially by integrating well-established research in the peer-reviewed literature in the introduction and discussion, such as studies that mirror what the authors present at the international level using within state variation in the US and bias analyses of existing case control studies on the relation between guns and suicide . The straw man use of the NRA detracts from an otherwise sober and thoughtfully written manuscript. Better to cite prior work that has built the case to date and to which this paper adds value.

Some of these papers include:

Azrael D and Miller M. Reducing Access to Lethal Means. A Review of the Evidence Base. Chapter XXVI. The International Handbook of Suicide Prevention, Second Edition. Edited by Rory C. O’Connor and Jane Pirkis. Published 2016 by John Wiley & Sons, Ltd.. Baffins Lane, Chichester, West Sussex PO19 1UD, England.

Miller M, Swanson S, Azrael D. Are We Missing Something Pertinent? A Bias Analysis of Unmeasured Confounding in the Firearm-Suicide Literature. Epidemiol Rev (2016) 38 (1): 62-69.doi: 10.1093/epirev/mxv011.

Swanson S, Eyllon M, Sheu Y, Miller M. Firearm access and adolescent suicide risk: Toward a clearer understanding of effect size. Injury Prevention. Published Online First: 14 May 2020. doi: 10.1136/injuryprev-2019-043605.

Miller M, Barber C, Azrael D, White R. Firearms and suicide in the United States: is risk independent of underlying suicidal behavior? Am J Epidemiol 2013 Sep 15;178(6):946-55. doi: 10.1093/aje/kwt197. Epub 2013 Aug 23.

In the discussion the authors focus only on the ecologic literature when making the case that firearms causally contribute to suicide rates. The argument could be strengthened by also referring to the individual-level case control and cohort studies that exist. I’m not sure that the argument they make when evoking reviews of the effect of legislation advances their argument. The authors should review those paragraphs and ask themselves what if anything does this text support of relevance to their important but modest point.

6. PLOS authors have the option to publish the peer review history of their article (what does this mean?). If published, this will include your full peer review and any attached files.

Reviewer #1: No

Reviewer #2: No

Reviewer #3: No

---

## [Author Response · Author response to Decision Letter 0]

10 Dec 2023

Revision Summary

The authors completely agreed with the recommendations. In general we added discussion of other relevant reports in the medical literature, four of which were selected, and removed the discussion on legislative impact and its 3 references). In total the manuscript was actually thus shortened, from 3,375 to 3,378 text words, and the number of references (45) being the same. Also, the figures and supporting information are being separately submitted as tiff files.

Reviewer #1:

The authors deserve my sincere congratulations. It has been quite some time since I reviewed such a good paper. The introduction is straightforward , the hypothesis is highly relevant, the data source is reliable, the statistical analysis is simple but efficient, and the discussion is clear and concise.

Reviewer #2:

The manuscript is sound and addresses an important topic. Statistical analysis is also valid. However, it lacks a literature review on the topic.

As also recommended by Reviewer #3, a literature review has been added:

“Although the primary limitation of this study is its ecologic nature, case control and cohort studies support firearm access per se as a causal factor of firearm mortality and not mental health per se. In the U.S., higher rates of firearm ownership at the state level have been shown to be strongly associated with higher rates of firearm suicide but not with non-firearm suicide or gun ownership level.[New reference39] The authors concluded that firearm ownership rates independent of underlying rates of suicidal behavior largely determine variations in suicide mortality across the 50 states.[New reference 39] In the U.S., a nationally representative study of 10,123 13-18 year-olds estimated that their risk of suicide was increased 3-4 times if they had lived in homes with a firearm compared with if they had not.[New reference 40] An analysis of four U.S. studies did not suggest any other confounders that explain the association between firearms and suicide.[New reference 41]

Other examples have evidenced that suicide rates can be substantially reduced without targeting underlying mental health or suicidality. The success in preventing suicides in Sri Lanka by reducing access to the most highly toxic pesticides is, as the authors conclude, “one of the strongest empirical arguments” and supports ‘household firearm ownership as a consistent strong predictor of suicide risk in studies that examined individual-level data.[New reference 42]” 

New References

39. Miller M, Barber C, White RA, Azrael D. Firearms and suicide in the United States: is risk independent of underlying suicidal behavior? Am J Epidemiol. 2013 Sep 15;178(6):946-55. doi:10.1093/aje/kwt197. 

40. Swanson SA, Eyllon M, Sheu YH, Miller M. Firearm access and adolescent suicide risk: toward a clearer understanding of effect size. Inj Prev. 2020 May 14;27(3):264–70. doi:10.1136/injuryprev-2019-043605. 

41. Miller M, Swanson SA, Azrael D. Are we missing something pertinent? A bias analysis of unmeasured confounding in the firearm-suicide literature. Epidemiol Rev. 2016;38(1):62-9. doi:10.1093/epirev/mxv011.

42: Azrael D and Miller M. Reducing suicide without affecting underlying mental health. Theoretical underpinnings and a review of the evidence base linking the availability of lethal means and suicide. Chapter 35. The International Handbook of Suicide Prevention, 2nd Edition. Edited by Rory C. O’Connor and Jane Pirkis. Published 2016 by John Wiley & Sons, Ltd.. Baffins Lane, Chichester, West Sussex PO19 1UD, England.

Authors should also rephrase some sentences to incorporate passive voice only. 

All active voice phrases, which were limited to the Methods section have been rephrased in the passive mode:

“The mean and 95% confidence interval (C.I.) for mental health disorder were obtained for the U.S. and for other high sociodemographic index (SDI)4 countries from the Institute of Health Metrics and Evaluation (IHME) Global Health Burden resource.5 For each of nine mental health disorders provided by IHME, the U.S. was compared with each of 40 other high SDI countries for overall firearm deaths, firearm homicides, and firearm suicides. Annual firearm death rate data in the U.S. was obtained from the Centers for Disease Control and Prevention Web-based Injury Statistics Query, Reporting and WONDER Systems.6,7 U.S. data were available until 2021 whereas IHME SDI country data were not available after 2019. The Joinpoint Regression Program, version 5.0.1-April 20238 and its constant variance option, was used to identify mortality trends and when they occurred. Joinpoint-derived p-values in the figures are designated in the text as <<0.001 if several logs <0.001.”

Reviewer #3:

The manuscript under review, “Fallacy of Attributing the U.S. Firearm Mortality Epidemic to Mental Health” is a cogent empirical essay that underscores an observation in need of reinforcement. Namely, that mental disorder prevalence in the U.S. is similar in all major categories to its 40 comparable sociodemographic countries and cannot explain the U.S. firearm crisis. None of this minimizes the need for the U.S. to support overall mental health, as the authors take care to note.

The paper could be strengthened substantially by integrating well-established research in the peer-reviewed literature in the introduction and discussion, such as studies that mirror what the authors present at the international level using within state variation in the US and bias analyses of existing case control studies on the relation between guns and suicide . The straw man use of the NRA detracts from an otherwise sober and thoughtfully written manuscript. Better to cite prior work that has built the case to date and to which this paper adds value.

Some of these papers include:

Azrael D and Miller M. Reducing Access to Lethal Means. A Review of the Evidence Base. Chapter XXVI. The International Handbook of Suicide Prevention, Second Edition. Edited by Rory C. O’Connor and Jane Pirkis. Published 2016 by John Wiley & Sons, Ltd.. Baffins Lane, Chichester, West Sussex PO19 1UD, England.

This may not be the correct reference information since the authors have the following chapter (#36 and not XXVI) in the 2nd edition of the book: Reducing Suicide Without Affecting Underlying Mental Health. Theoretical Underpinnings and a Review of the Evidence Base Linking the Availability of Lethal Means and Suicide. This chapter is now cited as summarized below.

Miller M, Swanson S, Azrael D. Are We Missing Something Pertinent? A Bias Analysis of Unmeasured Confounding in the Firearm-Suicide Literature. Epidemiol Rev (2016) 38 (1): 62-69.doi: 10.1093/epirev/mxv011.

Swanson S, Eyllon M, Sheu Y, Miller M. Firearm access and adolescent suicide risk: Toward a clearer understanding of effect size. Injury Prevention. Published Online First: 14 May 2020. doi: 10.1136/injuryprev-2019-043605.

Miller M, Barber C, Azrael D, White R. Firearms and suicide in the United States: is risk independent of underlying suicidal behavior? Am J Epidemiol 2013 Sep 15;178(6):946-55. doi: 10.1093/aje/kwt197. Epub 2013 Aug 23.

All four of these references (new #39-42) have been added to the Discussion as summarized below.

In the discussion the authors focus only on the ecologic literature when making the case that firearms causally contribute to suicide rates. The argument could be strengthened by also referring to the individual-level case control and cohort studies that exist. 

The references and additional comments in the Discussion: 

“Although the primary limitation of this study is its ecologic nature, case control and cohort studies support firearm access per se as a causal factor of firearm mortality and not mental health per se. In the U.S., higher rates of firearm ownership at the state level have been shown to be strongly associated with higher rates of firearm suicide but not with non-firearm suicide or gun ownership level.[New reference 39] The authors concluded that firearm ownership rates independent of underlying rates of suicidal behavior largely determine variations in suicide mortality across the 50 states.[New reference 39] In the U.S., a nationally representative study of 10,123 13-18 year-olds estimated that their risk of suicide was increased 3-4 times if they had lived in homes with a firearm compared with if they had not.[New reference 40] An analysis of four U.S. studies did not suggest any other confounders that explain the association between firearms and suicide.[New reference 41]

Other examples have evidenced that suicide rates can be substantially reduced without targeting underlying mental health or suicidality. The success in preventing suicides in Sri Lanka by reducing access to the most highly toxic pesticides is, as the authors conclude, “one of the strongest empirical arguments” and supports ‘household firearm ownership as a consistent strong predictor of suicide risk in studies that examined individual-level data.[New reference 42]” 

New References

39. Miller M, Barber C, White RA, Azrael D. Firearms and suicide in the United States: is risk independent of underlying suicidal behavior? Am J Epidemiol. 2013 Sep 15;178(6):946-55. doi:10.1093/aje/kwt197. 

40. Swanson SA, Eyllon M, Sheu YH, Miller M. Firearm access and adolescent suicide risk: toward a clearer understanding of effect size. Inj Prev. 2020 May 14;27(3):264–70. doi:10.1136/injuryprev-2019-043605. 

41. Miller M, Swanson SA, Azrael D. Are we missing something pertinent? A bias analysis of unmeasured confounding in the firearm-suicide literature. Epidemiol Rev. 2016;38(1):62-9. doi:10.1093/epirev/mxv011.

42: Azrael D and Miller M. Reducing suicide without affecting underlying mental health. Theoretical underpinnings and a review of the evidence base linking the availability of lethal means and suicide. Chapter 35. The International Handbook of Suicide Prevention, 2nd Edition. Edited by Rory C. O’Connor and Jane Pirkis. Published 2016 by John Wiley & Sons, Ltd.. Baffins Lane, Chichester, West Sussex PO19 1UD, England.

I’m not sure that the argument they make when evoking reviews of the effect of legislation advances their argument. The authors should review those paragraphs and ask themselves what if anything does this text support of relevance to their important but modest point.

This Discussion and its 3 associated references regarding legislative advances have been deleted.

---

## [Decision Letter · Decision Letter 1]

20 Dec 2023

Fallacy of Attributing the U.S. Firearm Mortality Epidemic to Mental Health

PONE-D-23-22215R1

Dear Dr. Bleyer,

We’re pleased to inform you that your manuscript has been judged scientifically suitable for publication and will be formally accepted for publication once it meets all outstanding technical requirements.

Kind regards,

Claudio Alberto Dávila-Cervantes, Ph.D.

Academic Editor

PLOS ONE

**NOTE FROM EDITORIAL STAFF: **In a previous round of review, one or more of the reviewers has recommended that you cite specific previously published works, and these citations have now been incorporated into the manuscript. As always, we recommend that you please review and evaluate the requested works to determine whether they are relevant and should be cited. It is not a requirement to cite these works and removing these articles will not affect this acceptance decision. We appreciate your attention to this request.

Reviewers' comments:

Reviewer's Responses to Questions

**Comments to the Author**

1. If the authors have adequately addressed your comments raised in a previous round of review and you feel that this manuscript is now acceptable for publication, you may indicate that here to bypass the “Comments to the Author” section, enter your conflict of interest statement in the “Confidential to Editor” section, and submit your "Accept" recommendation.

Reviewer #3: All comments have been addressed

2. Is the manuscript technically sound, and do the data support the conclusions?

Reviewer #3: Yes

3. Has the statistical analysis been performed appropriately and rigorously? 

Reviewer #3: Yes

4. Have the authors made all data underlying the findings in their manuscript fully available?

Reviewer #3: Yes

5. Is the manuscript presented in an intelligible fashion and written in standard English?

Reviewer #3: Yes

6. Review Comments to the Author

Reviewer #3: The authors have addressed all my concerns. This is a well written, well conceived and much needed paper.

7. PLOS authors have the option to publish the peer review history of their article (what does this mean?). If published, this will include your full peer review and any attached files.

Reviewer #3: No
